# Pharmacological Activation of Piezo1 Channels Enhances Astrocyte–Neuron Communication via NMDA Receptors in the Murine Neocortex

**DOI:** 10.3390/ijms25073994

**Published:** 2024-04-03

**Authors:** Andrea Csemer, Cintia Sokvári, Baneen Maamrah, László Szabó, Kristóf Korpás, Krisztina Pocsai, Balázs Pál

**Affiliations:** 1Department of Physiology, Faculty of Medicine, University of Debrecen, H-4002 Debrecen, Hungary; csemer.andrea@med.unideb.hu (A.C.); sokvari.cintia@med.unideb.hu (C.S.); baneen.maamrah@med.unideb.hu (B.M.); korpas.kristof@med.unideb.hu (K.K.); deak-pocsai.krisztina@med.unideb.hu (K.P.); 2Doctoral School of Molecular Medicine, University of Debrecen, H-4012 Debrecen, Hungary; laszlo.szabo@med.unideb.hu; 3HUN-REN DE Cell Physiology Research Group, H-4032 Debrecen, Hungary

**Keywords:** slow inward current, NMDA receptor, astrocyte, Piezo1, Yoda1, Dooku1, neocortex, pyramidal cell, glutamate

## Abstract

The Piezo1 mechanosensitive ion channel is abundant on several elements of the central nervous system including astrocytes. It has been already demonstrated that activation of these channels is able to elicit calcium waves on astrocytes, which contributes to the release of gliotransmitters. Astrocyte- and N-methyl-D-aspartate (NMDA) receptor-dependent slow inward currents (SICs) are hallmarks of astrocyte–neuron communication. These currents are triggered by glutamate released as gliotransmitter, which in turn activates neuronal NMDA receptors responsible for this inward current having slower kinetics than any synaptic events. In this project, we aimed to investigate whether Piezo1 activation and inhibition is able to alter spontaneous SIC activity of murine neocortical pyramidal neurons. When the Piezo1 opener Yoda1 was applied, the SIC frequency and the charge transfer by these events in a minute time was significantly increased. These changes were prevented by treating the preparations with the NMDA receptor inhibitor D-AP5. Furthermore, Yoda1 did not alter the spontaneous EPSC frequency and amplitude when SICs were absent. The Piezo1 inhibitor Dooku1 effectively reverted the actions of Yoda1 and decreased the rise time of SICs when applied alone. In conclusion, activation of Piezo1 channels is able to alter astrocyte–neuron communication. Via enhancement of SIC activity, astrocytic Piezo1 channels have the capacity to determine neuronal excitability.

## 1. Introduction

Piezo channels are mechanosensitive cation channels discovered in 2010 [1]. There are two members of this channel family, named Piezo1 and 2. Piezo2 was only described on neurons related to mechanosensation [2,3,4], whereas Piezo1 is abundant on several peripheral tissues and the central nervous system (CNS) [5]. In the periphery, it is present on the bone and cartilage [6,7], on muscles [5,8], on the cardiovascular system, importantly on endothelial cells [9,10] and the urinary system [11,12].

These channels are also found in almost all components of the central nervous system, including neurons, astrocytes, oligodendroglia, microglia and vessels [4]. Neuronal Piezo1 channels are responsible for developmental actions such as axonal growth and synaptogenesis [13,14,15]. Although the channel has a known contribution to long-term synaptic plasticity and cognitive functions [16], no direct actions on neuronal excitability have been demonstrated.

Astrocytes have a high density of Piezo1 channels. It is well demonstrated that activation of this channel by mechanical or pharmacological stimuli increases astrocytic calcium wave activity, which in turn increases gliotransmitter release from them [17,18].

Glutamate is not only the most abundant neurotransmitter of the CNS, but is also known as a gliotransmitter. Besides releasing glutamate, astrocytes regulate the concentrations of the ambient glutamate [19,20,21]. Ambient glutamate has its receptors on both neurons and glia. On neurons, extrasynaptic NMDA-, α-amino-3-hydroxy-5-methyl-4-isoxazolepropionic acid (AMPA-) and metabotropic glutamate (mGlu) receptors exist with various functions [22,23,24,25]. NMDA receptors can be responsible for tonic inward currents which generate a long-lasting increase in neuronal excitability. Depending on its release and diffusion characteristics, astrocytic glutamate release can lead to a transient increase in neuronal activity by ‘slow inward currents’ (SICs) [26,27]. These currents might synchronize the excitability of neighboring neurons or elicit timing-dependent synaptic plasticity [28]. Thus, these currents might have physiological actions on the fine tuning of neuronal excitability and synaptic strength, or pathological actions by helping the propagation of the spreading depression [29].

Although there is a growing body of evidence that astrocytes are regulated by the Piezo1 mechanosensitive channel, it has not been demonstrated whether it affects astrocyte–neuron communication. In this project, we recorded SICs on neocortical pyramidal cells to assess whether pharmacological activation or inhibition of Piezo1 channels affect this aspect of astrocyte–neuron communication. We found that the activation of the Piezo1 channel by its activator Yoda1 increases neuronal SIC frequency via NMDA receptors. Although the Piezo1 inhibitor Dooku1 did not significantly affect SIC frequency under control conditions, it altered the kinetics of the events when applied alone. Furthermore, it fully reverted actions of the Piezo1 activation. In conclusion, the mechanosensitive Piezo1 channel might be a powerful tool for regulation of astrocyte–neuron communication and neuronal activity.

## 2. Results

In the first set of experiments, we tested whether the pharmacological activation of Piezo1 channels is capable of changing the SIC activity on randomly chosen pyramidal neurons from layer V of the barrel field and trunk region of the primary somatosensory cortex, the medial and lateral parietal association cortex and the posterior area of the parietal cortex (Figure 1). SICs were distinguished from spontaneous excitatory postsynaptic potentials (sEPSCs) on the basis of their rise time using 20 ms as a cutoff value [28]. The shortest detected rise time of SICs was 33 ms, and the average was 211.68 ± 30.27 ms in contrast to the rise time of sEPSCs (longest rise time: 9.48 ms, average rise time: 7.56 ± 0.36 ms, *p* < 0.001). We performed all recordings in nominally magnesium-free aCSF in order to remove the magnesium blockade on NMDA receptors and to achieve a greater SIC frequency for better comparison of situations before and after the pharmacological manipulations of Piezo1 channels (Figure 2A–D). The addition of the Piezo1 activator Yoda1 (10 µM) did not significantly increase SIC amplitude (37.7 ± 4.88 pA in control and 67.28 ± 27.57 pA in Yoda1; n.s.), the charge transfer (area) of the individual events (13.96 ± 2.98 pC in control and 18.68 ± 3.47 pC in Yoda1; n.s.), or rise time (143.24 ± 23.67 ms in control and 168.1 ± 17.41 ms in Yoda1; n.s.) and decay time (449.8 ± 87.53 ms in control and 452.39 ± 54.31 ms in Yoda1; n.s.; Figure 2E). The SIC frequency showed a significant, more than twofold (216 ± 60%) increase when Yoda1 was applied (0.4 ± 0.1/min in control and 1.07 ± 0.29/min in Yoda1; *p* = 0.0185; Figure 2E). Mostly, this significant increase in frequency but probably also the tendency for an increase in other SIC parameters resulted in a significant 3.96 ± 0.94-fold increase of the charge transfer by SICs in a minute (named as ‘SIC activity’; 6.18 ± 1.96 pC/min in control and 19.6 ± 5.74 pC/min in Yoda1; *p* = 0.0165; Student’s *t*-test; n = 20; Figure 2E).

In the next experiments, we tested whether the actions of Yoda1 took place via the involvement of NMDA receptors (as SICs are elicited by NMDA receptor activation). Testing this hypothesis, we administered the NMDA receptor inhibitor D-AP5 and observed that SICs were abolished by this drug (n = 9; Figure 3A,B,D). Supplementation of the recording solution with Yoda1 did not cause any significant increase in SIC activity, as only a single SIC appeared on one of the neurons (SIC amplitude: 60.3 ± 38.48 pA in control; 0 in D-AP5 and 36.57 pA with additional Yoda1 by a single event, n.s.; charge transfer: 73.96 ± 65.41 pC in control; 0 in D-AP5 and 5.49 pC with additional Yoda1 by a single event, n.s.; rise time: 148.57 ± 44.19 ms in control; 0 in D-AP5 and 45.97 ms with additional Yoda1 by a single event, n.s.; decay time: 616.42 ± 340.62 ms in control; 0 in D-AP5 and 112.52 ms with additional Yoda1 by a single event, n.s.; SIC frequency: 0.27 ± 0.12/min in control; 0 in D-AP5 and 0.02 ± 0.02/min with additional Yoda1, *p* = 0.025 for control and D-AP5, Tukey’s multiple comparisons test; SIC activity: 13.21 ± 11.39 pC/min in control; 0 in D-AP5 and 0.13 ± 0.13 pC/min with additional Yoda1, n.s; Figure 3C,D).

As SICs are capable of influencing EPSC parameters [28], EPSCs recorded without SICs in the presence of D-AP5 were analyzed before and after application of Yoda1. The EPSC frequency with D-AP5 before Yoda1 application was 0.174 ± 0.122 Hz, whereas the amplitude was 10.43 ± 0.88 pA. With Yoda1 application, the frequency was 0.046 ± 0.01 Hz and the amplitude was 10.21 ± 0.59 pA, which did not differ significantly from the results obtained without Yoda1 (n = 7).

Next, we tested whether the Piezo1 inhibitor can reverse the increase in SIC activity elicited by pharmacological activation of the channel. At first, 10 µM Yoda1 was applied to induce the increase in SIC activity, which was followed by the addition of 10 µM Dooku1 (n = 13). As SICs can be influenced by several factors out of astrocytic Piezo1 channels [31], only those cases were considered where no SIC activity was detected under control conditions (Figure 4A–C; n = 8). After application of Yoda1, SICs appeared in all cases with a variable frequency (0.86 ± 0.18/min) and activity (15.3 ± 9.05 pC/min). Additional application of Dooku1 significantly diminished SIC frequency (0.053 ± 0.03/min; *p* < 0.0001) and reduced SIC activity (0.625 ± 0.511 pC/min; *p* = 0.0009; Figure 4D).

In the whole population of neurons where 10 µM Yoda1 and 10 µM Dooku1 were applied together, similar tendencies were observed as seen in the population where SICs were absent in the control and 10 µM Dooku1 was applied to reverse the actions of Yoda1 (n = 15). However, a decrease in SIC frequency and activity by application of Dooku1 was statistically not significant, likely because of the high variability of SIC parameters after Dooku1 application (SIC frequency: 0.27 ± 0.12/min in control, 0.96 ± 0.27/min in Yoda1, 0.36 ± 0.25/min with additional Dooku1; SIC activity: 3.95 ± 1.95 pC/min in control, 14.17 ± 3.76 pC/min in Yoda1, 6.49 ± 3.5 pC/min with additional Dooku1). In all 13 cases, the SIC frequency was reduced by additional Dooku1 (24.5 ± 7.8% of the data in Yoda1). In two cases out of thirteen, a mild increase in SIC activity was recorded with additional Dooku1 (1.3 and 27.6%), but a great reduction in SIC activity was seen on average (26.6 ± 10.4% of the data in Yoda1). In accordance with the previous experiments, other SIC parameters did not change significantly (SIC amplitude: 36.98 ± 5.57 pA in control, 79.39 ± 36.65 pA in Yoda1, 37.18 ± 8.49 pA with additional Dooku1; charge transfer: 24.07 ± 13.97 pC in control, 30.67 ± 10.7 pC in Yoda1, 22.33 ± 7.37 pC with additional Dooku1; rise time: 163.7 ± 33.86 ms in control, 262.13 ± 57.73 ms in Yoda1, 263.7 ± 108.67 ms with additional Dooku1; decay time: 405.98 ± 107.43 ms in control, 746.39 ± 286.5 ms in Yoda1, 765.58 ± 202.13 ms with additional Dooku1).

The experiment described above was repeated using a higher concentration of Dooku1 (20 µM). Only those cases were considered where SICs occurred under control conditions to assess whether the SIC activity of the initially active cells with further pharmacological stimulation of Piezo1 channels could be reverted by this drug (Figure 5A–C; n = 9). We found that, as expected, the SIC frequency and activity was significantly increased by Yoda1, which was effectively reverted by the higher concentration of Dooku1 (SIC frequency: 0.28 ± 0.09/min in control, 0.87 ± 0.23/min in Yoda1, 0.17 ± 0.07/min with additional Dooku1; SIC activity: 2.98 ± 0.89 pC/min in control, 10.91 ± 3.47 pC/min in Yoda1, 1.42 ± 0.54 pC/min with additional Dooku1). In all nine cases, the SIC frequency (11 ± 6% of the data in Yoda1) and activity (14.1 ± 7% of the data in Yoda1) were decreased by Dooku1 (Figure 5D).

Other SIC parameters did not show significant alterations (SIC amplitude: 34.73 ± 7.57 pA in control, 31.11 ± 6.39 pA in Yoda1, 28.61 ± 3.86 pA with additional Dooku1; charge transfer: 14.71 ± 3.84 pC in control, 16.63 ± 4.57 pC in Yoda1, 9.99 ± 2.66 pC with additional Dooku1; rise time: 226.56 ± 57.74 ms in control, 192.57 ± 40.14 ms in Yoda1, 141.38 ± 50.36 ms with additional Dooku1; decay time: 490.78 ± 112.82 ms in control, 446.59 ± 101.25 ms in Yoda1, 444.05 ± 196.28 ms with additional Dooku1).

The actions of the Piezo1 inhibitor Dooku1 (10 µM) were also tested to assess whether there is basic Piezo1 channel activity under control conditions (n = 9). When Dooku1 was applied alone, no significant difference was seen in the SIC amplitude (40.54 ± 5.94 pA in control, 49.08 ± 6.17 pA in Dooku1), charge transfer of the individual events (28.2 ± 5.8 pC in control, 22.33 ± 4.3 pC in Dooku1), the decay time (642.36 ± 167.24 ms in control, 477.38 ± 23.39 ms in Dooku1), the frequency (0.3 ± 0.12/min in control, 0.19 ± 0.08/min in Dooku1) and the SIC activity (6.98 ± 2.86 pC/min in control, 4.1 ± 1.59 pC/min in Dooku1; Figure 6A–E). Intriguingly, a significant decrease in the rise time was explored (338.01 ± 56.89 ms in control, 155.61 ± 9.17 ms in Dooku1; *p* = 0.006; Student’s *t*-test; Figure 6B,D,E). These results indicate that there is minimal Piezo1 activity affecting SICs even under control conditions.

## 3. Discussion

In this project, we observed the actions of Piezo1 ion channel opening and inhibition on SICs, the hallmarks of astrocyte–neuron communication. We found that the Piezo1 channel opener Yoda1 is capable of increasing the SIC frequency and activity. Application of the NMDA receptor inhibitor D-AP5 prevented these changes, whereas the Piezo1 inhibitor Dooku1 successfully reverted it. Application of Dooku1 alone significantly reduced the rise time of SICs but did not change any other parameters.

SICs are neuronal events which are consequences of astrocytic activation. This activation can be achieved by several actions, including activation of neurotransmitter receptors or swelling and other mechanical stimuli. These stimuli result in an increase in astrocytic calcium wave activity, which leads to gliotransmitter release. One of the important gliotransmitters is the glutamate, which is capable of activating neuronal extrasynaptic NMDA receptors in the perisynaptic regions or opposing the astrocytic processes in somatodendritic location. Depending on the released amount of glutamate, the kinetics of its release and the tortuosity of the diffusion pathways, SICs might have different kinetics, or in extreme cases, can be replaced with tonic inward currents [32,33,34,35].

SICs can exert two possible actions on neuronal networks. First, they synchronize depolarization of neuronal somata in the same astrocytic domain; as in this case, SICs will appear simultaneously on two neighboring somata. Second, they can synchronize synaptic strength; therefore, SICs are capable of eliciting timing-dependent synaptic plasticity [20,22,23,31].

For investigating whether SICs are the subject of mechanical stimulation via Piezo1, we had to exclude the mechanosensitivity of other channels. Most importantly, NMDA receptors are known to be mechanosensitive [36,37]. Therefore, instead of mechanical stimulation, we used openers and inhibitors of the Piezo1 channel in order to exclude direct mechanical actions on extrasynaptic neuronal NMDA receptors. It is well known that Piezo1 is found on several elements of the central nervous system, including the neurons and astrocytes, the two partners forming the SICs. There is a growing body of evidence that neuronal Piezo1 channels are present on several neuronal types [4]. This channel has an outstanding role in neurogenesis, neuronal migration, dendritic and axonal growth and synaptogenesis [4,13,14,15,38,39]. Recent studies described the role of Piezo1 channels in long-term synaptic plasticity and cognitive functions [4,16,40]. However, as involvement of microglia [16] and astrocytes [40] were shown, these actions might not be fully considered as actions exclusively exerted on neurons. It was previously shown that long-term synaptic plasticity is also altered or mediated by astrocytes [41,42,43], and GluN2B subunits containing NMDA receptor-mediated SICs have a particular role in it [28]. Mechanical (transcranial ultrasound or magneto-acoustic) brain stimulation via Piezo1 activation led to an increase in expression of proteins involved in long-term potentiation including the GluN2B subunit [16].

As astrocyte-dependent SICs are capable of changing SIC frequency and amplitude, we measured actions on EPSC frequency and amplitude in those cases where SICs were not present (in the presence of an NMDAR inhibitor) and did not find any difference in EPSC parameters with the application of Yoda1. One can conclude that no acute and direct actions of Piezo1 openings on synaptic currents were found by us; therefore, non-neuronal elements of the CNS are the main targets of Piezo1 openers modulating neuronal excitability.

The Piezo1 channel is abundantly present on astrocytes [4] and its activation increases astrocytic calcium wave activity [18]. Similar to neurons, astrocytic Piezo1 channels have a developmental role directly influencing astrocytes [44] and acting on astrocyte–neuron interactions [45]. Besides its roles in neuroinflammation [18,46], astrocytic Piezo1 contributes to long-term synaptic plasticity and cognitive functions, as selective ablation of the ion channel led to deficits in synaptic plasticity and cognitive functions, whereas overexpression of it led to an improvement in these functions [17]. In neuronal responses to acute mechanical stimuli via Piezo1 channels, astrocytes are known to have an important role. In the event that neuronal Piezo1 channel expression is missing, neuronal responses are not strongly affected, whereas if astrocytic Piezo1 is missing, neuronal responses are significantly reduced [40].

We observed stimulatory actions of Piezo1 activation on SICs. Based on the followings, we argue for the actions of SICs on astrocytes which led to these changes. Our arguments are as follows: (1) No direct actions on synaptic currents were found by us or have been described, (2) astrocytic mechanosensitivity and Piezo1 expression is well known, (3) astrocytic calcium wave activity is stimulated by Piezo1 activation [18], and this activity is coupled with gliotransmitter release and the occurrence of SICs, (4) the Piezo1 opener Yoda1 could not exert any significant action in the presence of the NMDAR inhibitor D-AP5.

Dooku1 alone only affected the rise time, leading to faster kinetics. This finding might let us conclude that (1) in line with the recent literature, Dooku1 has actions on Piezo1 independent from Yoda1 [47,48], and (2) SICs elicited by mechanical actions on astrocytes are slower than spontaneous ones [31,35]. Stimulation of astrocytes via activation of Piezo1 channels (either by mechanical or pharmacological actions) in turn leads to glutamate release which might have different mechanisms than release via stimulation of astrocytic G-protein coupled neurotransmitter receptors. This theory is supported by findings where hypoosmotic environment or inhibition of glutamate uptake elicited slower SICs than the spontaneous ones [32,33,35].

A finding underlying the complex nature of astrocytic activation and the consequential appearance of neuronal SICs is that Piezo1 activation elicited SICs on those neurons where this phenomenon was not present under control conditions and these SICs were almost fully abolished by the Piezo1 antagonist. In contrast, if those cases were also considered where SICs were present under control conditions, an increased antagonist concentration was needed to significantly revert the increased SIC activity. Potential explanations of this finding are as follows: (1) SICs can be elicited by the swelling- and neurotransmitter receptor-related activation of astrocytes as well, and combinations of these activation patterns are seen together. (2) There is a “balancing” mechanism in SIC activity mostly due to the activation and inactivation of extrasynaptic NMDA receptors. If there is no SIC under control conditions, neuromodulatory actions increase SIC activity, but if SICs frequently occur in the control, the same neuromodulatory actions might exert inhibition on their appearance [35].

## 4. Materials and Methods

### 4.1. Chemicals, Solutions

For the electrophysiological experiments, normal artificial cerebrospinal fluid (naCSF) was used. Its composition was as follows (in mM): NaCl, 120; KCl, 2.5; NaHCO_3_, 26; glucose, 10; NaH_2_PO_4_, 1.25; myo-inositol, 3; sodium-pyruvate, 2; ascorbic acid, 0.5; CaCl_2_, 2; MgCl_2_, 1 for incubation of the slices and 0 for recording; pH 7.4. Preparation of brain slices was performed in aCSF with reduced Na^+^ concentration (low Na^+^ aCSF). In this solution, 95 mM NaCl was replaced with 130 mM sucrose and 60 mM glycerol to maintain the same osmolarity. In order to remove the magnesium blockade of the NMDA receptors, all recordings were performed in nominally magnesium-free naCSF. All chemicals were purchased from Sigma-Aldrich (St. Louis, MO, USA), unless stated otherwise.

### 4.2. Animals, Preparation

Fifteen- to thirty-day-old homozygous floxed-stop- tdTomato (B6;129S6-Gt(ROSA)26Sor^tm9(CAG-tdTomato)Hze/^J; Jax mice accession number: 007905) mice from both sexes were used as experimental animals (n = 24). The strain was purchased from Jackson Laboratories (Bar Harbor, ME, USA). The reason for employing this strain was to use mice without any differences in phenotype compared to the wild-type mice, but to have the same genetical background of the strains used in our previous studies on SICs [28,35,49]. Experiments on mice were in accordance with the appropriate international (EU Directive 2010/63/EU for animal experiments), national and institutional guidelines and law on the care of research animals by the Ethical Committee for Animal Research, University of Debrecen, Hungary (approval number: 19/2019/DEMÁB). The number of animals involved in the experiments was reduced as much as possible by the experimental design and consideration of statistical necessities. Animal welfare was ensured by enrichment of the environment and regular checks, and by minimizing any interventions involving pain.

In preparation, 200 µm-thick brain slices, including the parietal cortex, were cut in the coronal plane from the area 1.22–2.06 mm caudal from the bregma. For preparation, ice-cold low Na^+^ aCSF and a Microm HM 650V vibratome (Microm International GmbH, Walldorf, Germany) was used. After preparation, slices were incubated at 37 °C for 1 h in naCSF.

### 4.3. Ex Vivo Electrophysiology

Patch clamp experiments were performed on layer V pyramidal neurons in the barrel field and the trunk region of the primary somatosensory cortex, the medial and lateral parietal association cortex and the posterior area of the parietal cortex of both sides. Patch pipettes with 6–8 MΩ resistance and potassium gluconate-based pipette solution were used with the following composition in mM: K-gluconate, 120; NaCl, 5; 4-(2-hydroxyethyl)-1- piperazineethanesulfonic acid (HEPES), 10; Na_2_- phosphocreatinine, 10; ethylene glycol-bis(β-aminoethyl ether)-N,N,N′,N′-tetraacetic acid (EGTA), 2; CaCl_2_, 0.1; Mg-ATP, 5; Na_3_-GTP, 0.3; biocytin, 8; pH 7.3. The patch setup was composed of an Axopatch 200A amplifier and a Digidata 1440 interface (Molecular Devices, Union City, CA, USA); recordings were performed with Clampex 10.0 software (Molecular Devices, Union City, CA, USA; version 10.2.0.18). Clampfit 10.0 (Molecular Devices; Union City, CA, USA; version 10.2.0.18) and Synaptosoft MiniAnalysis (Synaptosoft, Decatur, GA, USA; version 6.0.7) software was used for data analysis. The series resistance was kept below 20 MΩ with less than 10% change during recording. Only stable seals with minimal leak currents were included in the analysis.

For detecting SICs, a gap-free voltage clamp protocol was used with −60 mV holding potential. Transient inward currents with rise time greater than 20 ms were considered as SICs, based on literature data and our previous experiments [26,28]. Yoda1 (2-[5-[[(2,6-dichlorophenyl)methyl]thio]-1,3,4-thiadiazol-2-yl]-pyrazine; ProbeChem Biochemicals Ltd., Shanghai, China) [5,47,50,51] and D-AP5 (nonspecific inhibitor, Tocris Cookson Ltd., Bristol, UK) [52] were used in 10 µM concentration. Dooku1 (2-[(2,6-Dichlorobenzyl)thio)-5-(1H-pyrrol-2-yl)-1,3,4-oxadiazole; Tocris Cookson Ltd., Bristol, UK) was administered in 10 and 20 µM [5,53]. At the end of all recordings, NBQX (10 µM), D-AP5 (10 µM), strychnine (1 µM), bicuculline (10 µM) was administered to eliminate SICs and sEPSCs to exclude events recorded that occurred without activation of neurotransmitter receptors.

### 4.4. Morphological Analysis

Post hoc morphological evaluation of the recorded neurons was achieved with biocytin labelling during patch clamp recordings. After fixation (4% paraformaldehyde in 0.1 M phosphate buffer; pH 7.4; 4 °C), samples were permeabilized with Tris buffered saline (in mM, Tris base, 8; Trisma HCl, 42; NaCl, 150; pH 7.4) supplemented with 0.1% Triton X-100 and 10% bovine serum (60 min). Visualization of the biocytin labelling was achieved with streptavidin-conjugated Alexa488 (1:300; Molecular Probes Inc., Eugene, OR, USA; 90 min). Tile scan images of labeled neurons were taken with confocal microscope (Zeiss LSM 510; Carl Zeiss AG), using 40x objective and 1 μm z stacks. The position of neuronal somata was drawn with NeuroLucida software (MBF Bioscience, Williston, VT, USA; version 5.65)

All data were represented as average ± SEM. D’Agostino and Pearson omnibus normality tests were performed to determine the normal distribution of datasets. Statistical comparison of two datasets was achieved with two sample Student’s *t*-test, whereas multiple comparisons were conducted by using Tukey’s multiple comparisons test. Changes were considered significant when *p* < 0.05.

## 5. Conclusions and Future Directions

In conclusion, astrocyte–neuron communication is affected by the Piezo1 channels. These channels are less active in the resting state but are still able to influence the kinetics of SICs. Taken together with our previous study on the capability of SICs eliciting synaptic plasticity, one might assume that Piezo1 activation affects long-term potentiation and cognitive functions at least partially via astrocytic activation and the consequential stimulation of neuronal extrasynaptic receptors by gliotransmitters. These results probably help us to understand how Piezo1 contribute to long-term plasticity via astrocytes.

Manipulation on astrocytic Piezo1 channels might have important therapeutic significance. Astrocytic Piezo1 channel activation is related to cognitive functions, and its activation of Piezo1 by transcranial magnetic stimulation was beneficial for slowing cognitive decline in a model of Alzheimer’s disease [16,17]. It was also recently demonstrated that astrocyte-dependent SICs are important in synaptic plasticity both in mice and in humans [28]. However, there is limited information about the physiological functions of human astrocytic Piezo1. Probably, the first question to address in the future is whether Piezo1 has the same importance in human cognitive functions as was revealed by using mouse models.

There are a few studies on human tissue underlining the pathophysiological roles of astrocytic Piezo1. In temporal lobe epilepsy, Piezo1 expression is increased on astrocytes and other glio-vascular elements in correlation with the pro-inflammatory biomarkers in of humans [54]. Overexpression of Piezo1 in human glioblastoma also correlates with the malignancy of the tumor and the severity of the peritumoral brain edema [55,56,57]. In these diseases—and probably also in cognitive problems—astrocytic Piezo1 might serve as a therapeutic target.

## Figures and Tables

**Figure 1 ijms-25-03994-f001:**
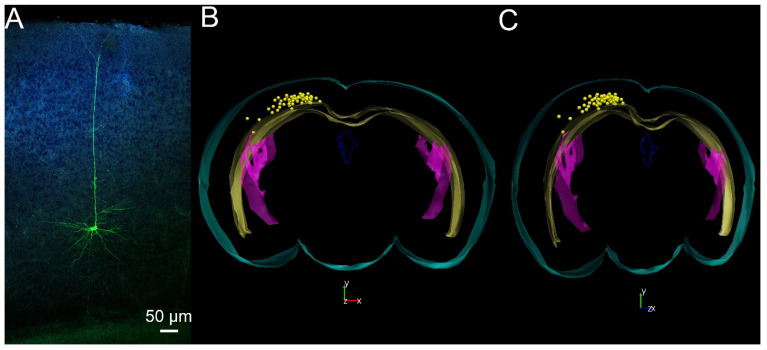
Morphological evaluation of the patched pyramidal neurons. (**A**) A layer V pyramidal neuron labelled with biocytin (green) supplemented with DAPI staining of the nuclei (blue). Scale bar: 50 µm. (**B**,**C**) Position of the somata from where SICs were recorded (somata: yellow spheres, light blue contour: brain surface, yellow contour: external capsule and corpus callosum, purple contour: lateral ventricle, blue contour: 3rd ventricle; (**B**) Frontal view, (**C**) 45° rotation). Drawing of the contours was based on the Paxinos atlas [30].

**Figure 2 ijms-25-03994-f002:**
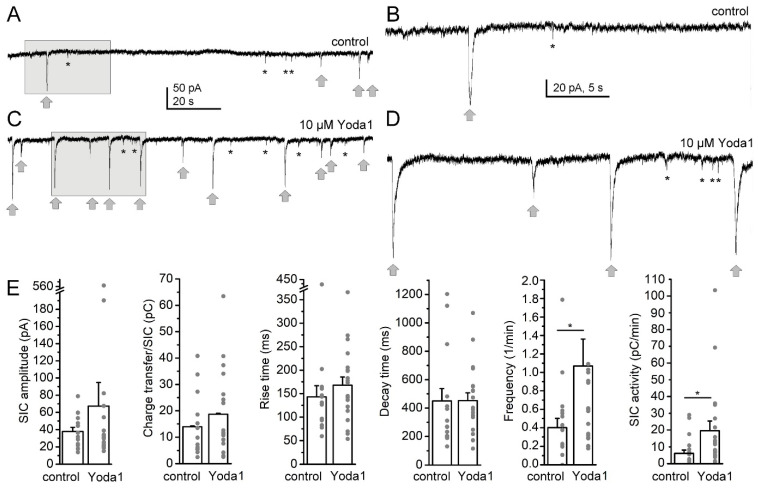
The Piezo1 opener Yoda1 increases SIC frequency. (**A**–**D**). Representative current traces under control conditions (in magnesium-free aCSF) and after addition of 10 µM Yoda1. (**A**). Current trace under control conditions. Gray arrows: SICs, asterisks: examples of EPSCs. Gray square: area magnified on panel (**B**). (**C**) Current trace with Yoda1. The area indicated with the gray square is magnified on panel (**D**)**.** (**E**) Statistical comparison of SIC parameters. The charge transfer is the area of the individual SICs, SIC activity is the area of all SICs in a minute. The columns and error bars represent average ± SEM, the gray dots are the individual datapoints. *: *p* < 0.05.

**Figure 3 ijms-25-03994-f003:**
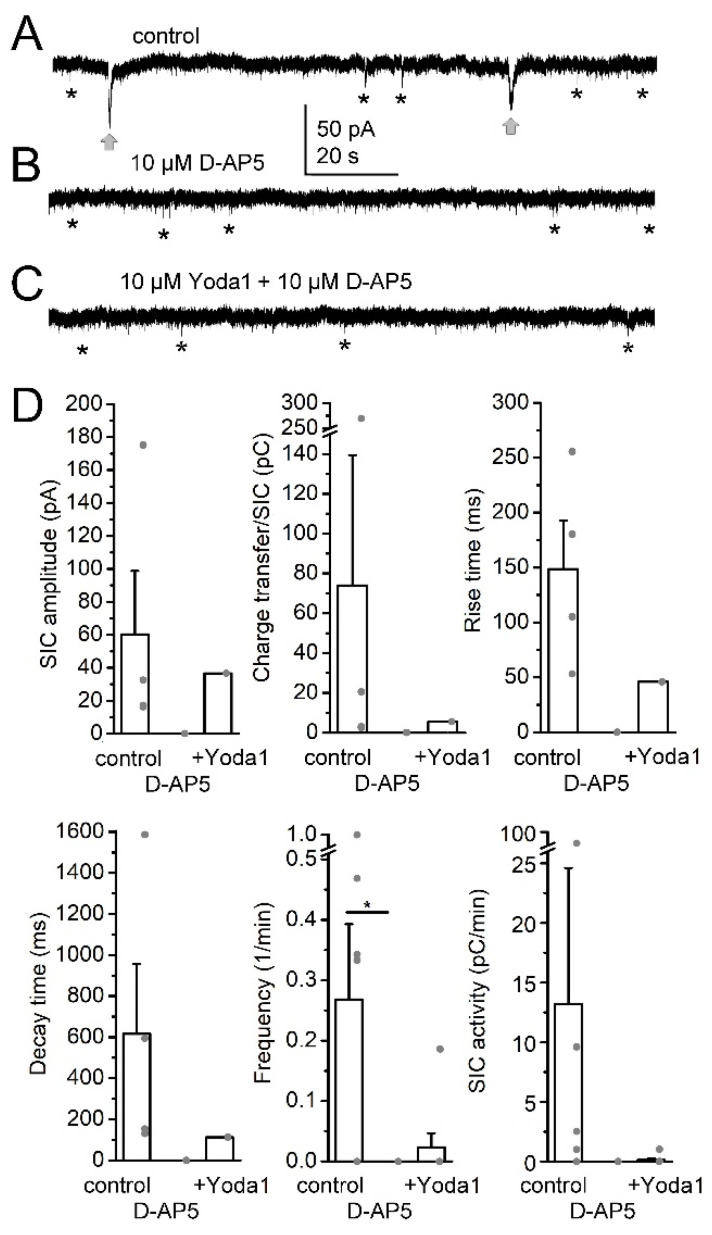
The NMDA receptor blocker D-AP5 prevents increase of SIC activity by Yoda1. (**A**–**C**). Representative current traces under control conditions (magnesium-free aCSF), 10 µM D-AP5 and additional 10 µM Yoda1, respectively (gray arrows: SICs, asterisks: examples of EPSCs). (**D**). Statistical comparison of SIC parameters with different treatments. The columns and error bars represent average ± SEM, the gray dots are the individual datapoints. *: *p* < 0.05.

**Figure 4 ijms-25-03994-f004:**
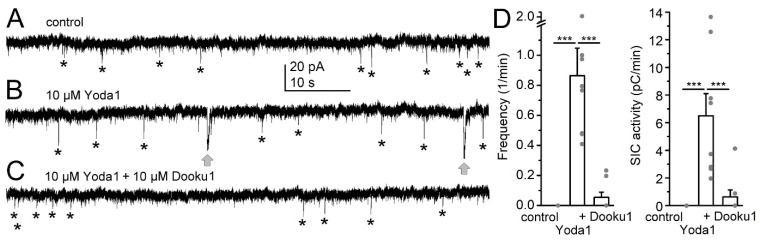
The Piezo1 inhibitor Dooku1 reverted the actions of Yoda1 on SICs. (**A**–**C**). Representative current traces under control conditions where no SICs appeared (magnesium-free aCSF, (**A**)), with addition of 10 µM Yoda1 (**B**) and supplementation with 10 µM Dooku1 (**C**). Gray arrows: SICs, asterisks: examples of EPSCs. (**D**). Statistical comparison of SIC parameters when 10 µM Dooku1 was used for reverting Yoda1 actions. The columns and error bars represent average ± SEM, the gray dots are the individual datapoints. ***: *p* < 0.001.

**Figure 5 ijms-25-03994-f005:**
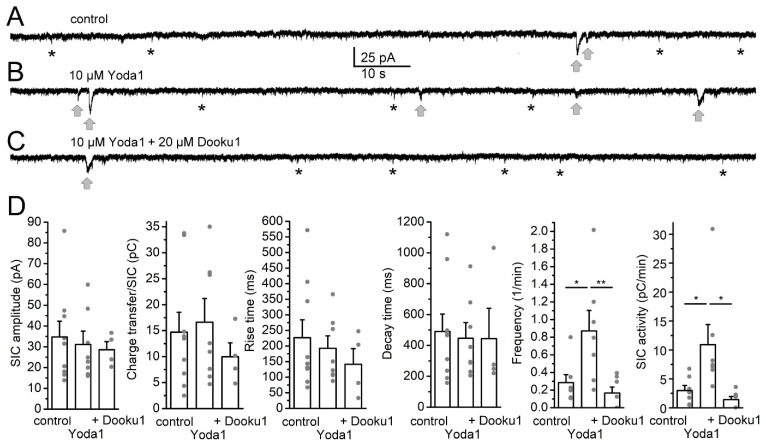
The Piezo1 inhibitor Dooku1 fully reverted the actions of Yoda1 on SICs in higher concentration. (**A**–**C**)**.** Representative current traces under control conditions (magnesium-free aCSF, (**A**)), with addition of 10 µM Yoda1 (**B**) and further addition of 20 µM Dooku1 (**C**). Gray arrows: SICs, asterisks: examples of EPSCs. (**D**). Statistical comparison of SIC parameters when 20 µM Dooku1 was used for reverting Yoda1 actions. The columns and error bars represent average ± SEM, the gray dots are the individual datapoints. *: *p* < 0.05; **: *p* < 0.01 (Tukey’s multiple comparisons test).

**Figure 6 ijms-25-03994-f006:**
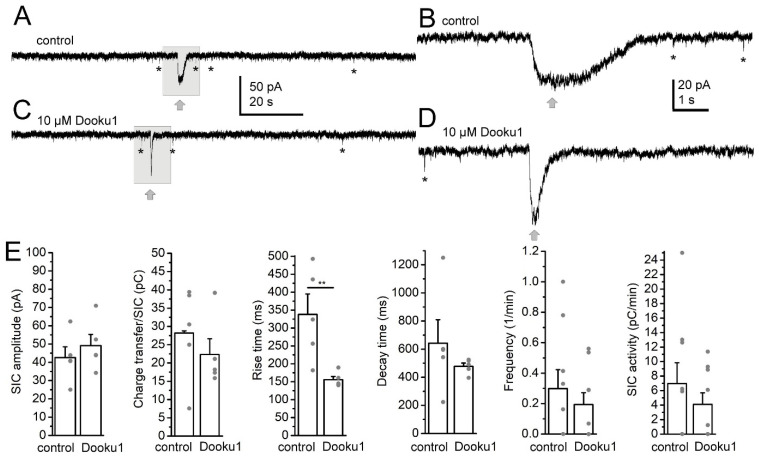
The Piezo1 inhibitor alone Dooku1 affected SIC rise time. (**A**–**D**). Representative current traces under control conditions (in magnesium-free aCSF, (**A**,**B**)) and with 10 µM Dooku1 (**C**,**D**). Gray arrows: SICs, asterisks: examples of EPSCs. Gray squares of B and D: areas magnified on panels A and C, respectively. (**E**). Statistical comparison of SIC parameters under control conditions and with Dooku1. Note that the rise time decreased with Dooku1 treatment (**B**,**D**,**E**). The columns and error bars represent average ± SEM, the gray dots are the individual datapoints. **: *p* < 0.01.

## Data Availability

Data are contained within the article.

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
