# Peer review of "Pharmacological Activation of Piezo1 Channels Enhances Astrocyte–Neuron Communication via NMDA Receptors in the Murine Neocortex"

_ijms, 2024, doi:10.3390/ijms25073994_

Round 1
Reviewer 1 Report
Comments and Suggestions for Authors
In this work, the authors elucidated the influence of the Piezo1 channel on the activity of slow inward currents of neurons. The text of the article is easy to read, the results are presented in detail. Some text transformations are required. After the corrections are made, the article can be published.
1) ‘NMDA’ (line 12), ‘NMDA-, AMPA-’ (line 47): Decipher the abbreviations (N-methyl-D-aspartate will not occupy much space of the Abstract).
2) Discussion (from the line 199): Conclude Discussion with prospects for further research and practical application of the findings.
3) Materials and Methods (from the line 286): For better understanding, divide this chapter into subchapters.
4) ‘Normality tests were performed to determine the normal distribution of datasets’ (lines 342-343): What specific tests have been used?
5) ‘Pál B. On the functions of astrocyte-mediated neuronal slow inward currents. Neur Regen Res., 2024; in press. https://doi.org/10.4103/ 431’ (lines 430-431): You cannot cite articles that have not yet been published (though you are the author) as references.
Author Response
Reviewer 1.
“In this work, the authors elucidated the influence of the Piezo1 channel on the activity of slow inward currents of neurons. The text of the article is easy to read, the results are presented in detail. Some text transformations are required. After the corrections are made, the article can be published.”
We are grateful to the Reviewer for the valuable comments and suggestions. We hope that we managed to address all of them sufficiently. All changes done in the manuscript are indicated with red font color.
- “‘NMDA’(line 12), ‘NMDA-, AMPA-’ (line 47): Decipher the abbreviations (N-methyl-D-aspartate will not occupy much space of the Abstract).”
The acronyms were deciphered where they first appeared in the text (lines 32, 67-68).
- Discussion(from the line 199): Conclude Discussion with prospects for further research and practical application of the findings.
We added a paragraph to the last chapter („Conclusions and future directions”), see lines 351-364.
„Manipulation on astrocytic Piezo1 channels might have important therapeutic significance. Astrocytic Piezo1 channel activation is related to cognitive functions and its activation on of Piezo1 by transcranial magnetic stimulation was beneficial to slower the cognitive decline in a model of Alzheimer’s disease [16-17]. It was also recently demonstrated that astrocyte-dependent SICs are important in synaptic plasticity both in mice and human [28]. However, there is limited information about the physiological functions of human astrocytic Piezo1. Probably the first question to address in the future whether Piezo1 has the same importance in human cognitive functions as it was revealed by using mouse models.
There are a few studies on human tissue underlining the pathophysiological roles of astrocytic Piezo1. In temporal lobe epilepsy, Piezo1 expression is increased on astrocytes and other glio-vascular elements in correlation with the pro-inflammatory biomarkers in of humans [53]. Overexpression of Piezo1 in human glioblastoma also correlates with the malignancy of the tumor and the severity of the peritumoral brain edema [54-56]. In these diseases –and probably also in cognitive problems-, astrocytic Piezo1 might serve as a therapeutic target.”
3) Materials and Methods (from the line 286): For better understanding, divide this chapter into subchapters.
The following subchapters were generated: Chemicals, solutions (line 270); Animals, preparation (line 281), Ex vivo electrophysiology (line 300), Morphological analysis (line 326)
4) ‘Normality tests were performed to determine the normal distribution of datasets’ (lines 342-343): What specific tests have been used?
We added the following sentence: „D'Agostino and Pearson omnibus normality tests were performed to determine the normal distribution of datasets.” (lines 336-337)
5) ‘Pál B. On the functions of astrocyte-mediated neuronal slow inward currents. Neur Regen Res., 2024; in press. https://doi.org/10.4103/ 431’ (lines 430-431): You cannot cite articles that have not yet been published (though you are the author) as references.
The cited paper became available online. The citation was corrected in line with the suggestions of the publisher.
Reviewer 2 Report
Comments and Suggestions for Authors
Dear the authors,
Csemer et al. examined in the current study that Piezo1 channels enhance astrocyte-neuron communication via NMDA-receptor in the murine neocortex. Basically, the manuscript is very interesting. However, the authors should address some questions. The comments are listed as below.
1. In the electrophysiology experiments using mouse slices, the authors stated that the measurements were done in the parietal cortex layers III-IV. The authors should show as a map-like plot where the measurements were actually done.
2. None of the observations made with the confocal microscope for various evaluations are shown. They should be shown.
3. The reviewer understand that this is a pharmacological experiment, but it was not clear whether piezo1 was truly expressed or activated at the locations measured. How did the authors confirm this?
4. The SEM are quite scattered, but are they really statistically significant? What is the reason for such variation?
Author Response
Reviewer 2.
Dear the authors,
Csemer et al. examined in the current study that Piezo1 channels enhance astrocyte-neuron communication via NMDA-receptor in the murine neocortex. Basically, the manuscript is very interesting. However, the authors should address some questions. The comments are listed as below.
We are grateful to the Reviewer for the valuable comments and suggestions on out manuscript. We especially acknowledge that a confusing typo was identified in the Materials and methods and we had the possibility to correct it (“layer V pyramidal cells instead of layer III-IV).
All changes done in the manuscript are indicated with red font color.
- In the electrophysiology experiments using mouse slices, the authors stated that the measurements were done in the parietal cortex layers III-IV. The authors should show as a map-like plot where the measurements were actually done.
- None of the observations made with the confocal microscope for various evaluations are shown. They should be shown.
In the present article, layer V pyramidal cells were involved. In the first version of the manuscript, we have not realized this confusing mistake. It was corrected in the present version.
We added a fluorescent image of a recorded pyramidal neuron and prepared a plot of the recordings. These images are found as Figure 1.
The corresponding parts of the Results (lines 91-92) and Materials and Methods (lines 295-296; 301-303) were also corrected.
For his help provided in confocal microscopy, László Szabó was added to the author list.
- The reviewer understand that this is a pharmacological experiment, but it was not clear whether piezo1 was truly expressed or activated at the locations measured. How did the authors confirm this?
Our present study is based on the pharmacological actions of a Piezo1 activator and an inhibitor. We showed that there are significant changes of astrocyte-dependent events but no change on synaptic events. The presence of Piezo1 on neuronal and non-neuronal structures was not investigated by us with morphological and molecular biological methods; as under our present conditions we did not have the possibility to perform these experiments. This information was based on literature data we cited.
Shi Z, Innes-Gold S, Cohen AE. Membrane tension propagation couples axon growth and collateral branching. Sci Adv. 2022; 8(35): eabo1297. doi: 10.1126/sciadv.abo1297.
Chu F, Tan R, Wang X, Zhou X, Ma R, Ma X, Li Y, Liu R, Zhang C, Liu X, Yin T, Liu Z. Transcranial magneto-acoustic stimulation attenuates synaptic plasticity impairment through the activation of Piezo1 in Alzheimer's disease mouse model. Research (Wash D C). 2023; 6: 0130. doi: 10.34133/research.0130.
Chi S, Cui Y, Wang H, Jiang J, Zhang T, Sun S, Zhou Z, Zhong Y, Xiao B. Astrocytic Piezo1-mediated mechanotransduction determines adult neurogenesis and cognitive functions. Neuron. 2022; 110(18): 2984-2999.e8. doi: 10.1016/j.neuron.2022.07.010.
Velasco-Estevez M, Rolle SO, Mampay M, Dev KK, Sheridan GK. Piezo1 regulates calcium oscillations and cytokine release from astrocytes. Glia. 2020; 68(1): 145-160. doi: 10.1002/glia.23709.
Zong B, Yu F, Zhang X, Pang Y, Zhao W, Sun P, Li L. Mechanosensitive Piezo1 channel in physiology and pathophysiology of the central nervous system. Ageing Res Rev. 2023; 90: 102026. doi: 10.1016/j.arr.2023.102026.
- The SEM are quite scattered, but are they really statistically significant? What is the reason for such variation?
Briefly, slow inward currents are mediated by astrocytic glutamate release preceded by astrocytic activation. This glutamate has a complex diffusion pathway with high tortuosity (unlike in the synaptic cleft). Changes of the cleavage of extrasynaptic glutamate can also vary SIC parameters. Relative positions of the astrocytic processes and the neuronal membranes facing to them are variable as well. All of these factors make SIC frequency, amplitude and kinetical parameters variable. Several mechanisms altering these factors will influence SICs (via different ways of glutamate release, clearance, diffusion or NMDA receptor inactivation) and factors stimulating SICs can be inhibitory on them under different background conditions (Kovács and Pál, 2017). I recently summarized properties of SICs and factors influencing them in a review paper (Pál B. On the functions of astrocyte-mediated neuronal slow inward currents, Neural Regen. Res. 19(0)000-000; doi: 10.4103/NRR.NRR-D-23-01723). The reason of this variability was mentioned in the manuscript, as well as the statistical tests and the p values calculated with the experiments in question.
Reviewer 3 Report
Comments and Suggestions for Authors
To enhance the clarity and flow of your paper, it's recommended to reorganize the structure as follows: Introduction, Materials and Methods, Results, Discussion
Methods
For a more comprehensive evaluation of the ethical considerations, it would be beneficial for the paper to include specific details about the ethical approval process, such as the name of the ethics committee that approved the research and the approval number.
Additionally, it should describe the steps taken to minimize animal suffering and to adhere to the principles of the 3Rs (Replacement, Reduction, and Refinement) – fundamental ethical guidelines in the use of animals in research.
Discussion
Implications for Treatment: The discussion could be expanded to more directly address the implications of these findings for the treatment and understanding of neurological diseases.
Future Directions: Including a section on future research directions would not only highlight the potential of this field but also how this study contributes to the broader research landscape.
Author Response
Reviewer 3.
To enhance the clarity and flow of your paper, it's recommended to reorganize the structure as follows: Introduction, Materials and Methods, Results, Discussion
We are gratefult to the Reviewer for the positive comments which helped us to make the manuscript better. We hope that we managed to address all concerns related to the manuscript.
All changes done in the manuscript are indicated with red font color.
The organization of the manuscript was not changed as we followed the instructions of the journal to authors and their recommendation is to have the Materials and Methods after the Discussion.
Methods
For a more comprehensive evaluation of the ethical considerations, it would be beneficial for the paper to include specific details about the ethical approval process, such as the name of the ethics committee that approved the research and the approval number.
Additionally, it should describe the steps taken to minimize animal suffering and to adhere to the principles of the 3Rs (Replacement, Reduction, and Refinement) – fundamental ethical guidelines in the use of animals in research.
Following the suggestions above, we made the following changes of the Materials and Methods (lines 290-294):
„Experiments on mice were in accordance with the appropriate international (EU Directive 2010/63/EU for animal experiments), national and institutional guidelines and law on the care of research animals by the Ethical Committee for Animal Research, University of Debrecen, Hungary (approval number: 19/2019/DEMÁB). The number of animals involved in the experiments was reduced as much as possible by the experimental design and consideration of statistical necessities. Animal welfare was ensured by enrichment of the environment and regular checks; and by minimizing any interventions involving pain.”
Discussion
Implications for Treatment: The discussion could be expanded to more directly address the implications of these findings for the treatment and understanding of neurological diseases.
Future Directions: Including a section on future research directions would not only highlight the potential of this field but also how this study contributes to the broader research landscape.
We added the following paragraph (lines 351-364) and citations 53-56 to the „Conclusions” chapter. The chapter was renamed to „Conclusions and future directions":
„Manipulation on astrocytic Piezo1 channels might have important therapeutic significance. Astrocytic Piezo1 channel activation is related to cognitive functions and its activation on of Piezo1 by transcranial magnetic stimulation was beneficial to slower the cognitive decline in a model of Alzheimer’s disease [16-17]. It was also recently demonstrated that astrocyte-dependent SICs are important in synaptic plasticity both in mice and human [28]. However, there is limited information about the physiological functions of human astrocytic Piezo1. Probably the first question to address in the future whether Piezo1 has the same importance in human cognitive functions as it was revealed by using mouse models.
There are a few studies on human tissue underlining the pathophysiological roles of astrocytic Piezo1. In temporal lobe epilepsy, Piezo1 expression is increased on astrocytes and other glio-vascular elements in correlation with the pro-inflammatory biomarkers in of humans [53]. Overexpression of Piezo1 in human glioblastoma also correlates with the malignancy of the tumor and the severity of the peritumoral brain edema [54-56]. In these diseases –and probably also in cognitive problems-, astrocytic Piezo1 might serve as a therapeutic target.”
Round 2
Reviewer 2 Report
Comments and Suggestions for Authors
None
Reviewer 3 Report
Comments and Suggestions for Authors
The authors responded appropriately to the reviewers' review.
Congratulations on completing your thesis.